# Lithium Harvesting from the Most Abundant Primary and Secondary Sources: A Comparative Study on Conventional and Membrane Technologies

**DOI:** 10.3390/membranes12040373

**Published:** 2022-03-29

**Authors:** Fraz Saeed Butt, Allana Lewis, Ting Chen, Nurul A. Mazlan, Xiuming Wei, Jasmeen Hayer, Siyu Chen, Jilong Han, Yaohao Yang, Shuiqing Yang, Yi Huang

**Affiliations:** 1School of Engineering, Institute for Materials & Processes, The University of Edinburgh, Robert Stevenson Road, Edinburgh EH9 3FB, UK; f.s.butt@sms.ed.ac.uk (F.S.B.); a.m.lewis-6@sms.ed.ac.uk (A.L.); t.chen-38@sms.ed.ac.uk (T.C.); n.a.mazlan@sms.ed.ac.uk (N.A.M.); x.wei-13@sms.ed.ac.uk (X.W.); s1685255@sms.ed.ac.uk (J.H.); s.chen-206@sms.ed.ac.uk (S.C.); 2School of Chemical and Pharmaceutical Engineering, Hebei University of Science and Technology, Shijiazhuang 051432, China; 3Jiangsu Dingying New Materials Co., Ltd., Changzhou 213031, China; yaohao.dingyingmaterials@gmail.com (Y.Y.); dingyingmaterial@163.com (S.Y.)

**Keywords:** lithium, lithium recovery and recycling, membrane technologies, li-rich brines, lithium-ion batteries

## Abstract

The exponential rise in lithium demand over the last decade, as one of the largest sources for energy storage in terms of lithium-ion batteries (LIBs), has posed a great threat to the existing lithium supply and demand balance. The current methodologies available for lithium extraction, separation and recovery, both from primary (brines/seawater) and secondary (LIBs) sources, suffer not only at the hands of excessive use of chemicals but complicated, time-consuming and environmentally detrimental design procedures. Researchers across the world are working to review and update the available technologies for lithium harvesting in terms of their economic and feasibility analysis. Following its excessive consumption of sustainable energy resources, its demand has risen sharply and therefore requires urgent attention. In this paper, different available methodologies for lithium extraction and recycling from the most abundant primary and secondary lithium resources have been reviewed and compared. This review also includes the prospects of using membrane technology as a promising replacement for conventional methods.

## 1. Introduction

Lithium (Li), the 25th most abundant element on earth, is found in two isotopes; ^6^Li (7.59% population) and ^7^Li (92.41% population) [1]. In 2015 there was estimated to be 20 mg kg^−1^ of Li embodied in the earth’s crust—present in over 150 minerals [2]. Li is also found in continental brines, geothermal waters and seawater [3,4,5,6,7] which have been reported to have a greater abundance of Li than the hard rock equivalent [4]. In recent decades, Li requirement has increased dramatically from ~2000 tonnes in 2005 to >14,000 tonnes in 2020 [8,9]. It is also estimated that by 2025, global lithium consumption would have more than doubled from current levels. Consequently, lithium prices have hit a historical high of £13,000/tonne, reflective of the ever-increasing global energy demand. In the predictable future, lithium would continue to be one of the most in-demand commodities in the world and its environmental impacts associated with energy and material recovery are considered controversial [10,11].

Historically, lithium has been used for the production of glassware and ceramic material, greases, lubricants and rubbers, lightweight alloys, polymers, air treatment, rocket propellant industries, vitamin A and, of course, in pharmaceuticals (Figure 1) [9]. This unique and lightweight alkali metal has an outstanding electrochemical potential of 3.04 V, a high energy density and excellent conductive abilities along with long-life expectancies [12]. For instance, Liu et al. recently reviewed the performance of various batteries and found that lithium-ion batteries had many obvious advantages in comparison to lead-acid, nickel-cadmium, nickel-metal hybrids, and redox flow cells [13]. Furthermore, the use of lithium batteries is predicted to produce better hybrid and electric-powered vehicles [14,15,16,17,18,19] as well as demonstrate great potential for large capacity green energy storage [20].

In terms of abundance, seawater brines (59%) and mineral clays (25%) are the most profound naturally occurring primary sources of lithium, with seawater brines dominating the natural supply (Figure 2a) [21]. However, Li does not occur naturally in its free state due to its highly reactive nature, hence more stable compounds such as Li_2_CO_3_, LiOH or LiCl are generally formed. More importantly, in different resources, they normally co-exist with abundant other ions including, but not limited to, magnesium, calcium, iron, sodium, potassium, borates, sulphate, and bicarbonates, which makes lithium harvesting much more challenging [22,23,24,25]. Among these resources, lithium recovery from lithium-bearing minerals and clays (spodumene, lepidolite, zinnwalidite, ambloygonite and petalite) has been well studied. Some commonly used methods developed to date include chemical leaching [26], bioleaching [27], and pressure leaching [28]. Whilst harvesting a high purity Li_2_CO_3_ at 99%, these conventional processes are generally energy-intensive and cause environmental concerns [29,30]. For example, lithium derived from Portuguese granite rock is around 2.5 times more costly than lithium collected from Chilean brine reserves. Hence, owing to the high availability of the aqueous reservoirs, such sources can serve as a major supply for effective lithium recovery in comparison to their hard rock equivalents (Figure 2a).

Furthermore, lithium recovery and recycling from secondary resources has quickly grown in importance to accommodate the ever-rising demand for lithium consumption through sustainable lithium harvesting. Over the past few years, out of all the available secondary resources, lithium-ion batteries have emerged as the most prominent source for lithium recycling, accounting for 35% of total lithium consumption which is expected to be doubled over the next decade (Figure 2b) [31]. For instance, the electrification of the global transport sector is in demand for lithium-ion batteries and some countries (e.g., Norway) are leading the way in sustainable, circular battery production. As more LIBs are demanded, it becomes even more significant to recycle and reuse them. Although LIBs contain a reasonable percentage of Li (5–7 wt.%), only 3% of the total spent LIBs are recycled, with minimal focus on lithium recycling [21]. However, there has been growing and remarkable attention on the development of sustainable lithium recycling technologies from used lithium-ion batteries.

In this review, methodologies developed recently for lithium extraction and recycling from the most abundant primary and secondary lithium resources (continental brines and LIBs, respectively) are thoroughly reviewed. A direct comparison between conventional technologies and membrane-based lithium harvesting methods is drawn, focusing on strengths and weaknesses within the existing processes. A special focus will be given to membrane-based technologies being sought out to offer systemic approaches to tackle the above-mentioned technological challenges [2]. Owing to the small footprint, good treatment effect, and low cost, membranes have received increasing attention in the precious metal recovery field.

## 2. Methodologies for Liquid-Based Lithium Harvesting and Their Environmental Impacts

A variety of techniques have been developed in the past decades for effective lithium extraction from aqueous sources. These methods include ion exchange (exchange of ions between the liquid and solid phase) [32], adsorption (transfer of components from liquid onto the solid surface) [33], solvent extraction [34], and precipitation [35] (Figure 3). The strengths and deficiencies of lithium harvesting both from seawater brines (Primary Li source) and LIBs (secondary Li source) using conventional technologies are summarized in Table 1. Compared to hard rock lithium mining (e.g., crushing, grinding and dense medium separations), lithium recovery from aqueous sources has received increasing attention. This is owing to the aqueous extractions being comparatively less energy-intensive and more cost-effective. However, the presence of Mg^2+^ in brines poses a challenge for effective and efficient separations due to the great similarity in chemical properties between Li and Mg ions. Tuning the performance of extraction technologies has become a key focus in recent studies in order to optimise Li extractions in complex Mg:Li ratio brines [36]. In addition to dealing with the complicated and time-consuming separations, these conventional techniques typically produce a large volume of waste and cause severe corrosion of the system [31,37,38,39,40].

Similarly, a range of methods have been proposed for lithium recycling from spent LIBs, including hydro-metallurgy [2], pyro-metallurgy [41], bio-metallurgy [42] and hybrid processes (Figure 3) [38,43,44,45]. The above-mentioned separation techniques are often assisted by several pre-treatment processes to facilitate an efficient lithium recovery. The as-received LIBs are first discharged by dipping them in salt solutions to avoid spontaneous combustion or short-circuiting. Later, the batteries are dismantled into different parts including plastic, electrodes and electrolyte before forwarding them for subsequent lithium extraction processes [46]. However, these pre-treatment processes cannot effectively address the problems associated with excessive hazardous chemical consumption and pose significant environmental issues. Furthermore, the secondary phase of the processes, Li recycling, is highly energy-intensive and time-consuming [46].

Among lithium harvesting technologies, membrane-based processes are a relatively novel technique. These processes offer many advantages compared with conventional methods, such as easy operation, low energy consumption, high efficiency, small footprints and ease of scalability [44,46]. Therefore, membrane-based processes are highly promising to act as a preferable technique for effective lithium recovery. In recent years, a wide range of membrane-based processes have been developed, particularly for lithium recovery from brines and seawater. Apart from typical pressure-driven membrane separation processes, such as nanofiltration (NF) [47], many integrated membrane-conventional methods and hybrid processes have also been reported, including membrane-electrodialysis [32], membrane-adsorption [48], and membrane-solvent extraction [49].

To meet the sharply growing Li demand and also to overcome the barriers of lithium harvesting from brines and lithium-ion batteries, more cost-effective, efficient and environmentally friendly techniques are highly demanded. In the meanwhile, the technological development and improvement of existing lithium mining and recycling processes are also of critical importance to promote a more sustainable Li future.

## 3. Lithium Recovery from Brines/Seawater

### 3.1. Conventional Methods

Despite intensive technological development, conventional extractions from brines have succumbed to too many deficiencies and require careful and systematic adaptations for varied brine conditions. In most recent research, these conventional techniques (i.e., precipitation, liquid-liquid extraction and adsorption) have been incorporated with membrane technologies to improve Li extracting efficiency and also to reduce the industrial carbon footprint, however, they have not been commonly practised in industry.

#### 3.1.1. Precipitation

In the 1990s, precipitation was primarily employed to precipitate Li after the removal of co-existing ions by applying solar evaporation and various precipitants (Equations (1)–(5)). Typically, this consists of seven stages (Figure 4) that alter the composition of the brine; concentrating or precipitating the brine at each stage. The main ions of interest for removal are Mg^2+^ and Ca^2+^ as the smaller ions (e.g., Na^+^) are readily extracted by evaporation via crystallization [48,49,50].
Mg^2+^ + Strong Alkali → Mg Carbonate or Mg Salt(1)
Mg^2+^ + Ca(OH)_2_ → Mg(OH)_2_ + Ca^2+^(2)
Ca^2+^ + CaCl_2_ + coexisting ions → CaSO_4_•2H_2_O(3)
2Li^+^ + Na_2_CO_3_ → Li_2_CO_3_ + 2Na^+^(4)
Mg^2+^ + Ca(OH)_2_ + SO_4_^2−^ + 2H_2_O → CaSO_4_•2H_2_O + Mg(OH)_2_(5)

A high Mg/Li ratio has been shown to have a negative effect on Li separation, although this has been improved over the years. Newer precipitation methods such as using layered double hydroxides (LDH) intercalated with the Mg had unveiled many other shortcomings, for instance, low Li recovery due to primary formation of LiAl_2_(OH)_6_Cl·xH_2_O [51,52]. Despite recent advancements, most precipitation-based processes are usually very time-consuming and produce significant amounts of waste.

#### 3.1.2. Solvent Extraction

Solvent extraction has been considered as an effective hydro-metallurgical separation technique and has demonstrated several technological strengths—a simple, continuous operation that is easily adaptable [53,54]. This process normally consists of four major stages, as shown in Figure 5, with the solvent being recirculated throughout the process and lithium removed as an extractant [55]. The solutes are induced into equilibrium with the organic solvent before scrubbing to remove the undesired solutes. The addition of HCl into the raffinate strips the mixture, replacing Li^+^ with H^+^, and the new mixture is then regenerated to restart the process [56]. A typical example of this method is using tributylphosphate (TBP)/Kerosene with FeCl_3_ as a co-extraction agent which requires low pH to avoid hydrolysis of ferric ions [55,57]. In this method, one of the challenges is the selection of suitable solvents, as common solvents have a preference for H^+^ rather than Li^+^ or a low attraction affinity for the solute. In addition, the development of a more efficient scrubbing stage is highly desirable. It has been found that in a continuous operation with multiple scrubbing stages aided by centrifugation, Li extraction rate has been improved significantly [58,59].

In recent studies, ionic liquids (ILs) were employed to improve the practicality of the process. They have attractive solvent extraction properties such as negligible volatility, nonflammability, high thermal and electrochemical stability, and outstanding ionic conductivity even under anhydrous conditions [60]. Previously, typical ILs such as hexafluorophosphate (${\mathrm{PF}}_{6}^{-}$) and bis(trifluoromethyl sulfonyl)imide (${\mathrm{NTf}}_{2}^{-}$) were employed due to their immiscibility in water. However, this results in fluoride hydrolysis to hydrofluoric acid (Equation (6)) [61].
(6)$$6H^{+}+{\;\mathrm{PF}}_{6}^{-}+6H_{2}O+{\mathrm{HNO}}_{3}\;\to \;H_{3}{\mathrm{PO}}_{4}+6\mathrm{HF}+{\mathrm{HNO}}_{3}+2H_{2}O$$

To eliminate the unfavourable products (HF), functionalized ionic liquids (FILs) which could promote the interactions between metal-coordinate groups and the metal ion solute were studied. The functionalization of the ionic liquid has previously been achieved using functional groups such as alkyls, phosphates or amino, for example [61,62]. More recently, phosphate-based FILs were employed and a lithium extraction efficiency of 70% was reported. Bai et al. performed a detailed study on the lithium separation mechanism using extraction and stripping for brines with large Mg/Li ratios. It has been found that the addition of trialkylmethylammonium di(2-ethylhexyl)orthophosphinate, tributyl phosphate (TBP) and FeCl_3_ in Mg-dense brines led to the formation of [Li·2TBP][FeCl_4_], which upon stripping resulted in the formation of lithium enriched complexes, i.e., Li.2TBP [63]. Overall, the application of FILs has achieved a high lithium selectivity, enabling fast absorbance and interference-free lithium extraction [61,62,63,64,65]. It has also been found that FILs and ILs have a lower energy barrier than solvent alone [58], however, they require a high pH condition [66].

To tackle these challenges, the application of synergistic solvent extraction has been extremely useful in amplifying lithium extractions [56]. Such solvents can be defined as having a greater extraction capability when working in combination rather than independently. Hence, this class of materials has been of great research interest in recent years and also has demonstrated great effectiveness in synergistic solvent extraction [56,57]. Interestingly, Zhang et al. determined that the amplification of synergic effect would be greater with alkyl groups in comparison to alkoxy groups. Furthermore, it has been found that synergic reagents such as TPPO (triphenylphosphine oxide) would reduce the synergic abilities within the mixture because of the conjugation of benzene rings with P=O, which decreases the electron density around the P=O. By optimizing the concentrations and interactions between two solvents, synergistic solvent extractions have achieved up to 90% Li extraction in natural and synthetic brines [59].

Although giving promising approaches for lithium harvesting, these solvent extraction methods usually produce a large volume of waste materials and require expensive co-agents to improve process efficiency. Moreover, TBP solvents are highly corrosive, which could cause severe damage to the primary equipment.

#### 3.1.3. Adsorption

The adsorption method has been considered as one of the most convenient technologies for lithium recovery from aqueous resources and is especially suitable for lithium recovery from salt lake brines with a high Mg/Li ratio and seawater brines with complex compositions [33]. Adsorption differs from the ion exchange process, and lithium ions are separated selectively from aqueous solutions through physical or chemical adsorption interactions. A flow chart for the lithium extraction via a typical adsorption process is displayed in Figure 6.

##### Lithium Ion-Sieve (LIS) Method

The lithium ion-sieve (LIS) method provides an effective approach to lithium recovery from solutions containing different ions and thus has been regarded as the most promising adsorption technology for lithium recovery from aqueous solution [67]. The lithium ion-sieve process can be described as the ‘‘LIS effect’’ [68]. In short, when the LIS adsorbent is placed in aqueous solutions, lithium ions are adsorbed prior to undergoing stripping from the adsorbent through a Li-H ion exchange process. Thus, leading to the exchange of Li^+^ with H^+^ inside the LIS structure. As lithium-ion has the smallest ionic radius among all metal ions, only lithium-ion itself can re-enter these sites. Therefore, LIS is placed in solutions containing different metal ions and highly efficient selective adsorption of lithium ions occurs.

##### Lithium Ion-Sieve Adsorbents (LISs)

Lithium ion-sieve adsorbents (LISs) refer to lithium selective adsorbents with unique chemical structures and properties which are capable of separating lithium effectively from briny aqueous resources [33]. They have the advantages of high lithium uptake capacity, excellent lithium selectivity, satisfactory recycle performance and an environmentally friendly lithium adsorption/desorption process. The existing LISs can be classified into two major types according to chemical elements: (i) the lithium manganese oxides-type (LMO-type) [68] and (ii) the lithium titanium oxides-type (LTO-type) [69]. Spinel LMO-type is the major type of lithium-ion sieve, and its lithium extraction follows a redox mechanism [70], ion exchange mechanism [71] or a combination of both. It has been well studied that the LMO-type LISs showed high lithium adsorption capacities, outstanding lithium selectivity and excellent regeneration performance, although the regeneration process could be expensive. However, the dissolution of Mn^2+^ during the regeneration process with acid degrades the ion exchange capacity and results in poor cycling stability and serious water pollution issues. This key issue seriously limits LMO-type LISs potential for upscaling. Therefore, the cost-effective, environmentally friendly and simple regeneration of spinel LMOs have been highly desirable [72].

Comparatively, the LTO-type LISs are environmentally friendly as the titanium compounds can be easily removed from an aqueous solution [33]. In addition, the LTO-type LISs have higher molecular stability due to the large titanium-oxygen bond energy. However, the large-scale industrial application of LTO-type LISs in lithium extraction from aqueous solution has been very limited due to inefficient lithium adsorption/desorption cycles. Furthermore, low lithium recovery efficiency has hindered their applications in the industry. To tackle these challenges, future works should be focused on the development of attractive LTO-type sorbents for selective lithium extraction with superior advantages including high ion-exchange capacity, high lithium selectivity, high stability and economic efficiency [73,74].

The lithium extraction via the ion-sieve adsorption process has the obvious advantages of simple operation and low energy consumption. It is particularly suitable for extracting lithium from the salt lake brines with high magnesium to lithium ratio. However, the recovery process requires a long contact time for lithium ions and the adsorbents. Moreover, the adsorbents used are usually powdery and expensive and may degrade during the acid-driven desorption process.

Despite their many advantages, conventional techniques have demonstrated various disadvantages such as high energy consumption, large waste production and excessive operational requirements (Table 1). In order to accomplish equivalently high yields and/or purity of Li demonstrated in conventional extraction methods from brines, and overcome their shortcomings, membrane technologies have been widely investigated (Table 2).

### 3.2. Membrane-Based Separation Processes

As afore-discussed, despite some promising and constructive progress made, conventional methods for lithium harvesting still suffer from numerous disadvantages, such as low efficiency, high energy consumption, and severe environmental concerns. In recent years, membrane-based separation technology has emerged as a promising alternative for lithium separation and is preferred over conventional techniques. There are various benefits of membrane methods that have been the focus of recent lithium separation research in the lithium sector due to its excellent selectivity. More specifically, membranes advance the separation in low concentrations of different species and have greater abilities to operate effectively in applications that demand purity. Other benefits for considering this lithium purification method are the simple membrane unit design, ease of operation, low-cost and simple installation technique.

#### 3.2.1. Nanofiltration

Nanofiltration (NF) is a pressure-driven membrane-based separation technology [75,76,77]. An NF membrane has a molecular weight cut-off of 0.2–10 kDa, between that of reverse osmosis (RO) (<0.2 kDa) and ultrafiltration (UF) (1–500 kDa) membranes. Hence, it has the unique capability of removing inorganic salts from salt aqueous solution [77]. Especially, NF membranes are capable of preferentially extracting monovalent ions from solutions containing multivalent ions.

The selective separation behaviour of NF membrane is based on two basic types of exclusion mechanisms: steric exclusion mechanism and charge based exclusion mechanisms [78]. The steric exclusion mechanism is the geometric exclusion of solute particles larger than the membrane pore size. As the pore size of an NF membrane is typically between 1 nm and 10 nm, particles/molecules with big size and high molecular weight, therefore, can be excluded from desired solutions. However, NF membranes usually have a slightly charged surface, and the dimensions of pores are close to the dimensions of ions. Therefore, the interactions between solutes and membrane cannot be governed by the steric hindrance alone but also relies on non-sieving rejection mechanisms [79], i.e., the charge-based exclusion mechanisms.

The charge-based exclusion mechanisms include dielectric exclusion (DE) [79] and Donnan exclusion [80]. The Donnan exclusion is due to the charged nature of the NF membrane, as well as the interactions of co-ions with fixed electric charges. Owing to the charge on the NF membrane, a natural repulsion of similarly charged ions will occur at the membrane surface. Comparatively, ions with opposing charges will be attracted to the membrane surface and be drawn through the membrane pores. Hence, when placed in a salt solution, a potential difference at the interphase is generated to counteract the transport of co-ions to the membrane as well as counter-ions to the bulk solution [81]. In this way, the co-ions are repelled from the membrane, and counter-ions are also rejected due to electroneutrality requirements; thus, salt as a whole is rejected. The dielectric exclusion (DE) results from interactions between ions and polarised interfaces of media with different dielectric constants [81]. The primary effect is caused by the difference between the two dielectric constants of the aqueous phase and the polymeric matrix. Hence, when an ion is situated in the media with a higher dielectric constant (e.g., water), it induces electric charges with the same sign as itself at the interface between the media with a lower dielectric constant (e.g., membrane) [81]. Thus, the exclusion of ions from membrane pores occurs. The diagram presented in Figure 7 schematically explains each of the exclusion mechanisms [75].

In recent years, NF has been extensively reported as an efficient approach to a range of industry challenges, including wastewater reclamation, dyes rejection, and the separation of monovalent ions from co-existing multivalent ions. In particular, NF membranes are found to be highly effective in terms of the recovery of lithium from lithium-containing aqueous solutions, such as brine or seawater. Somrani et al. investigated the separation of lithium from salty Tunisian lake brines using the NF membranes and low-pressure reverse osmosis (LPRO) membranes [82]. NF membranes appeared to be more successful in extracting Li^+^ from a diluted brine due to its higher hydraulic permeability to pure water, low critical pressure of zero Pa and higher monovalent ion selectivity that can be achieved at low working pressures (less than 15 bar). It was also found that NF membranes were preferable to LPRO membranes in terms of lithium-magnesium separation. Bi et al. also studied the recovery of lithium from high Mg^2+^/Li^+^ ratio brine by nanofiltration [77]. In their study, NF proved to be an efficient approach to recover Li^+^ and reduce the Mg^2+^/Li^+^ ratio from brines with a high Mg^2+^/Li^+^ ratio. They also proved that the mass transport inside the NF membrane is governed by the combination of steric hindrance, Donnan exclusion, and dielectric exclusion. Sun et al. studied the separation and enrichment of lithium from brine with a high Mg^2+^/Li^+^ ratio using a Desal (DL) NF membrane [83]. They found that a low pH benefited the separation by increasing the rejection rate of magnesium and decreasing the rejection rate of lithium, while a high Mg^2+^/Li^+^ ratio negatively affected the separation by increasing the rejection rate of lithium and decreasing the rejection rate of magnesium. Yang et al. filtrated the Mg^2+^/Li^+^/Cl^−^ solutions with a commercially available nanofiltration membrane to investigate the possibility of enriching the lithium component [44]. Within a certain concentration range, their studies found that the Mg^2+^/Li^+^ ratio and the Li^+^ concentration did not affect the separation factor. Wen et al. investigated the applicability of NF for recovering lithium chloride from lithium-containing solutions by performing a process assessment experiment. A diagram explaining the experimental process for NF treatment is presented in Figure 8 [84]. It was found that steric hindrance became remarkable at higher concentrations due to the formation of ion pairs, ion clusters, and molecules.

Commercially available NF membranes are mostly negatively charged. However, it has been recently discovered that the positively charged NF membranes tend to be more efficient for the separation and recovery of multivalent cations such as Mg^2+^ and Ca^2+^. This property is especially of great importance in terms of efficient lithium and magnesium separation. Therefore, the development of positively charged NF membranes for efficient lithium recovery has attracted general interest. Li et al. developed a positively charged polyamide composite nanofiltration hollow fibre membrane via the interfacial polymerization of 1,4-Bis(3-aminopropyl) piperazine (DAPP) and trimesoylchloride (TMC) on the polyacrylonitrile (PAN) ultrafiltration hollow fibre membrane [85]. The membrane was applied for lithium and magnesium separation, and after the filtration by the composite membrane, the mass ratio of Mg^2+^/Li^+^ decreased from an initial 20:1 to 7.7:1, in the MgCl_2_ and LiCl mixtures, respectively.

Zhang et al. fabricated a positively charged NF membrane through interfacial polymerization—with a polyethersulfone (PES) three-channel capillary UF membrane as the substrate and polyethyleneimine (PEI) as the aqueous precursor (Figure 9) [86]. The membrane exhibited long durability and good separation performance for Mg^2+^ and Li^+^ when applied to separate mixed salts solution simulating the composition of salt lake brine. Li et al. synthesized a composite NF membrane with a positively charged skin layer via interfacial polymerization between branched poly (ethylene imine) (BPEI) and trimesoyl chloride (TMC) with crosslinked polyetherimide as the support (Figure 10) [87]. The obtained membrane showed efficient lithium recovery performance from simulated brine (LiCl/MgCl_2_) with a separation factor Li/Mg of about 9.2:1 and displayed excellent durability for 36 h of filtration.

Overall, NF membranes have emerged as an efficient approach for lithium extraction from brines, and they have the advantage of providing high permeability with lower energy requirements while maintaining high rejection performance. However, NF technology suffers from limitations such as membrane fouling, insufficient separation, membrane lifetime and chemical resistance. Despite the challenges in direct lithium recovery from brines, it is suggested that NF technology would be highly advantageous in many other industrial processes, such as precipitation and evaporation [74].

#### 3.2.2. Membrane Solvent Extraction

Owing to the promising performance shown in solvent extraction (see Section 3.1.2 solvent extraction for more details), recent attention has been drawn to the fabrication of membranes which support such extractions. The membranes are used to promote the solvents ability to extract the desire materials, and hence reduce the volume of waste typically produced by solvent extraction alone. Creating a homogeneous interface, these operations use supported liquid membranes (SLMs) which have previously demonstrated high selectivity and low energy utilization [88,89]. SLMs have been the subject of many recent investigations for the separation of metal ions from industrial waste effluents using a variety of extractants. For example, they could act as ion exchange membranes for the lithium ions whilst blocking the organic solvent from passage to an aqueous solution [88]. In a recent study, successful lithium separation via SLMs has been achieved by complexation or binding with specific chemical species. Song et al. studied polyethersulfone (PES) and sulfonated poly-phenyl ether ketone (SPPESK) in the synthesis of hydrophilic nanoporous membranes as a stabilizing barrier for liquid-liquid membrane extraction of lithium ions. In this study, using tributylphosphate (TBP) as the extractant and kerosene as the diluent, lithium extraction and stripping were demonstrated in both single-staged and sandwiched membrane extraction contactor systems [88]. In their following studies, Song et al. further improved the stability of similar membranes, such as poly(ethylene-co-vinyl) (EVAL). The membrane structure provided good chemical resistance with reduced swelling (ethyl section) and created a hydrophilic domain for ion transportation (vinyl alcohol section) [90]. In this case, the lithium diffused from the brine solution towards the membrane interface and crossed over the swollen membrane. Upon arrival at the extraction interface, the lithium bonded with cationic compounds in the extractant fluid to form LiFeCl_4_ which released the previously attracted Na^+^ ion. This Na^+^ ion then passed through the membrane in the reverse mechanism as Li^+^. This entire process was driven by the concentration gradient in an osmosis mechanism (Figure 11) [90]. Overall, the results gave a linear correlation between the Li feed concentration and the concentration of extraction with the greater EVAL content, suppressing macro voids to provide a more compact structure. This is believed to be due to the unique properties of the materials.

Although these membranes have been considered as successful applicants for liquid-membrane extractions, SLMs still have some issues with stability, durability and solvent leakage. Conquering these shortcomings requires future research into the fabrication of the organic membranes whilst maintaining hydrophilicity to increase solvent resistance and reduce membrane swelling for reduced fouling [89].

#### 3.2.3. Membrane Adsorption

Membrane adsorption contributes to a large percentage of conventional-membrane hybrid techniques. This methodology has been adapted primarily using polymeric materials with surface enhancements [91,92] and has brought a wide array of materials to the market for Li adsorption. Chung et al. and Sun et al. demonstrated the use of various polymeric substrates such as poly(vinylidene fluoride) –(C_2_H_2_F_2_)_n_–, polyvinylpyrrolidone –(C_6_H_9_NO)_n_–, polysulfone, polyester, and Kimtex^®^ composites [38,39]. Furthermore, Lu et al. prepared various adsorption samples based upon a polyethersulfone substrate [93]. Meanwhile, Park et al. reported a polysulfone (PSf)-based mixed matrix nanofiber (MMN) [94]. Each of these materials has been widely used in membrane research due to its high porosity, pressure resistance, elasticity, and stability.

Hence, a wide array of polymeric substrates has been advantageous, providing a stable microporous material for hierarchical membrane fabrications. However, to selectively contain lithium, the pore size of the surface material was required to be on the nanoscale. This was accomplished by various research teams after the addition of an active layer onto the substrate to create an unsaturated chemical group that attracts and holds the lithium [93]. Lu et al. explained the cause of this effect was due to the synergic effects between the lithium and the hydroxyl groups within the structure [37]. Additionally, they suggested that the increased roughness of the materials improved the hydrophilicity and hence made this sample very useful in lithium harvesting from brines.

#### 3.2.4. Membrane Electrodialysis

Membrane-Electrodialysis has become a common practice for Li extractions over the past decades, with a wide array of applications in the industry [63] The efficiency of this electro-membrane separation process has been amplified in multiple cases by alteration of the membrane stack (a unit component within the electrodialysis system). Implementing various membranes, such as ion-exchange membranes, bipolar membranes and novel membranes technologies such as activated carbon have achieved excellent efficiency and, in some cases, reduced the energy demand [95].

##### Selective Electrodialysis

Recently, the employment of selective membranes, such as monovalent selective cation exchange membranes (CIMS) and monovalent selective anion exchange membranes (ACS), in electrodialysis has shown great potential in the effective separation of positively and negatively charged ions in aqueous mixtures. A typical setup contains 11 CIMS and 10 ACS sandwiched between a cathode and anode [96,97]. The system set-up (Figure 12) [97], can be further optimized for Li^+^ separation through voltage and temperature adjustment. A small change in the voltage (e.g., 3 V) has proven to boost the performance, giving a comparable 67.65%:80.08% Li recovery rate [98,99]. Similarly, a 20 °C difference in operating temperature has been shown to improve lithium recovery from 21.47% to 39.2%. Such voltage and temperature tuning are also dependent on specimen compositions. For example, ion composition in the feed has a vital effect on Li recovery efficiency. As reported, at elevated operating temperatures, increased recovery rates have been observed with Mg^2+^ and Ca^2+^ ions whilst a decreasing efficiency in the presence of Na^+^ [96].

Due to the wide variation in Li sources, such optimization is essential for feed streams with different ion compositions, making this process less adaptive and very time-consuming. Furthermore, the attraction of anions to the anode can result in the formation of dangerous gases, such as Cl_2_, which are hazardous and corrosive to the equipment [97,98].

##### Ion-Exchange Membranes

Ion-exchange membranes are the most commonly used membrane species in electrodialysis and have been applied widely across the research board with studies relating to Li recovery [100]. Liu et al. developed a new sandwiched liquid ion-exchange membrane, designed to selectively extract Li^+^ from brines with a high Mg/Li ratio [101]. The results showed excellent Li^+^ recognition and rapid electromigration of Li^+^ with the assistance of the electric field. At a current density of 5.437 gA m^−2^, the Mg/Li decreased from 50:1 in initial feeding brine to 0.5:1 in 12 h. Song and Zhao reported a hybrid method of ion-exchange membrane and precipitation for lithium recovery from Li_3_PO_4_ [37]. They found that Li and P were efficiently separated by cation-exchange membranes. P/Li mass ratio of the catholyte was reduced to 0.23, which is 6.5 times lower compared to the feed at 1.48. The lithium concentration of the purified catholyte solution was 22.5 g L^−1^. Guo et al. adopted a selective-electrodialysis method (S-ED equipped with monovalent selective ion-exchange membrane) to recover lithium from seawater/salt lake brines [98]. The seawater results showed, at higher voltage, the recovery of the ratio of Li^+^ improved but excessive-high working voltage would adversely affect the separation between Li^+^ and Mg^2+^. For the salt lake brines, the recovery ratio of Li^+^ was 76.45% with a specific energy consumption of 0.66 kWh.

Another study by Shi et al. designed a monovalent selective cation exchange membrane (CIMS) assembled in membrane capacitive deionization (MCDI) to separate lithium from magnesium [12]. These authors report a removal rate of Li^+^ and Mg^2+^ in large modules achieved was 38.4% and 19.2%, respectively. Even though the separations were not high, they managed to reduce energy consumption to 0.0018 kWh mol^−1^, which is lower than that of the electrodialysis range between 0.04–0.27 kWh mol^−1^. However, these membranes have also been reported for their ageing dynamics in the presence of various chemicals. For instance, studies show the spontaneous deterioration of the membranes in the presence of sodium hypochlorite (a common oxidizing agent used in reverse osmosis, ultrafiltration, and microfiltration) [102].

More recently, metal-organic frameworks (MOFs) have attracted great attention in academia and industry due to their versatile properties and remarkable potential for wide applications, including lithium recovery [103]. MOFs are organic-inorganic hybrid solids with infinite and uniform crystalline coordination networks [104]. Consisting of metal ions/clusters and organic linkers, these materials have proven promising for ion conduction and transportation [99]. Guo et al. reported an intergrown and continuous polystyrene sulfonate (PSS) threaded HKUST-1 membranes through an in situ confinement conversion process (Figure 13) [105]. The as-prepared PSS@HKUST-1-6.7 membrane has uniquely anchored three-dimensional sulfonate networks for ion transportation due to the linear polymer (PSS). As a result of the different size sieving effects and affinity differences of the Li^+^, Na^+^, K^+^, and Mg^2+^ ions to the sulfonate groups, the PSS@HKUST-1-6.7 membrane displayed ideal selectivity for Li^+^ over Na^+^, K^+^, and Mg^2+^ with binary ion separation factors of 35, 67, and 18 [15], respectively, which is the highest ever reported among ionic conductors and Li^+^ extraction membranes. Therefore, the membrane was considered a very promising material for the efficient extraction of lithium ions from salt-lake brines.

Another recent study by Zhang et al. reported a hybrid membrane of polyvinyl chloride matrix filled with MOFs (MOFs@PVC) for extraction of Li^+^ from salt-lake brines with high Mg^2+^/Li^+^ [106]. They employed six MOFs including ZIF-8, UiO-66, HSO_3_-UiO-66, HKUST-1, MOF-808, SO_4_-MOF-808 to fabricate MOFs@PVC via casting. The ion selection property was studied by the current-voltage (I-V) plots via two compartment transport cells as shown in Figure 14 [106]. They also reported HSO_3_-UiO-66@PVC membrane that showed highest selectivity for Li^+^ (Li^+^/Mg^2+^ > 4) with a diffusion coefficient of 2.0 × 10^−10^ cm^2^ s^−1^. The pore size and sulfonation of MOFs play important roles in the separation of Li^+^/Mg^2+^. The pore size provides pore channels for ion transportation and sulfonated groups anchored in the MOFs can delay Mg^2+^ transfer because of the strong affinity between sulfonated groups and Mg^2+^, which enhanced selectivity for lithium-ion.

##### Bipolar Membranes

Some cases of membrane adaption have led to the use of bipolar membranes to separate the acids and bases from a mixture. This has been advantageously applied to lithium extraction due to the aqueous nature of the feed solution [107]. By conjunction with bipolar membranes and ion-exchange membranes, Li is efficiently separated from existing co-ions as well as effectively separating boron in the same manner (Figure 15) [95].

Hwang et al. designed an enhanced bipolar membrane electro-dialysis (BEDI) to recover lithium ions from lithium manganese oxide (LMO) [108]. Three types of bipolar membranes modules were designed; bipolar membrane modules with 2 sheets, 3 sheets, and 4 sheets. The conditions for optimal lithium recovery such as pH, voltage and flow rates were evaluated. The authors revealed that at the optimum conditions when the number of bipolar membrane sheets was 4, under a pH lower than 4, a voltage of 6.5 V and a flow rate of 0.44 mL cm^−2^ min^−1^, the desorption efficiency of lithium was approximately 70%, with recovery time reduced by approximately 180 min compared to the chemical process.

Another type of bipolar membrane process for lithium and cobalt separation was bipolar membrane electrodialysis coupled with metal-ion chelation (EDTA) reported by Lizuka et al. [42]. The separation experiment was conducted using a three-cell type of electrodialysis system as shown in Figure 16 [42]. The electrodialysis unit consists of three cells divided by two bipolar membranes (BPM), one anion-exchange membrane (AEM), and one cation-exchange membrane (CEM). The cobalt ions were chelated by EDTA and lithium-ion was hardly chelated. The selectivity for each metal was approximately 99%.

## 4. Lithium Recovery from Lithium-Ion Battery

Lithium-ion batteries are becoming an integral part of renewable-based energy systems that helps to provide an efficient and greener solution for energy storage. LIBs have found their use in a variety of applications ranging from portable electronic devices to energy grid systems. Owing to the reduction in CO_2_ emission and improved energy to fuel weight ratio, LIBs have also been widely used in electronic vehicles. LIBs have been especially desirable in this case due to their high charge to mass potential in comparison to other battery types [109,110].

In the recent decade, the extensive use of LIBs has posed not only a great threat to the world’s lithium resource depletion but also the prevailing problem concerning the consumed and non- recycled LIBs. Hence, immediate attention to alleviate any danger to the ecosystems due to the release of harmful chemicals is required [109]. Currently, as low as 3% of LIBs are recycled [111]. In a report, “Recycling rates of metals” published by UNEP in 2011, less than 1% of lithium is being recycled from LIBs [112]. To maintain a balance between lithium supply and demand, proper management of lithium resources, the development of highly cost-efficient waste disposal techniques and proper documentation of the environmental safety regulations are highly desirable [111,112]. Recently, efforts have been made to upgrade the already existing technologies and the developing new methods for Li recovery from both primary and secondary sources. The main aspect of these studies is to improve the sustainability of existing recycling processes and maintain economical and industrial feasibility.

### 4.1. Conventional Methods

Currently, the commercial processes used for recycling and refining of lithium and other metals (including nickel, copper, cobalt, and aluminium) from LIBs can be divided into two major categories: (i) pre-treatment processes and (ii) metal-extraction processes [109].

#### 4.1.1. Pretreatment Process

In a typical pre-treatment process, the spent LIBs are firstly discharged using saturated-salt solutions (e.g., NaCl and Na_2_SO_4_ salt solution) to prevent short-circuiting or self-ignition caused by combustion [113]. Furthermore, it is recommended to recycle the electrolyte before the discharging stage. This is achieved by using organic solvent extraction or supercritical carbon dioxide to prevent the formation of hazardous vapours from electrolyte (LiPF_6_) and salt contact [114,115]. The use of supercritical carbon dioxide has proven to be more effective as it does not contaminate the electrolyte and the electrolyte recovery is significantly simplified [116,117,118]. Then, the obtained batteries are disassembled manually to separate the cathode from the anode to facilitate metal extraction and further processing [119]. Different solvents are in use to dissolve the organic binder to effectively separate the cathode from aluminium foil using the solvent dissolution method [120,121,122,123]. Zhou et al. have found 60 °C as an optimum temperature for effective removal of polyvinylidene fluoride (PVDF) binder through dissolution in dimethylformamide (DMF) [124]. Elsewhere, Zhang et al. used 15 vol% of trifluoroacetate (TFA) for dismantling the cathode from the aluminium foil through a solid-state reaction at relatively mild conditions of 40 °C for 180 min. The optimised liquid to solid (L/S) ratio was found to be 8 mL g^−1^ [125].

Another pre-treatment technique being used for the effective removal of strongly bonded PVDF from aluminium foil and the cathode material is ultrasonic-assisted separation [126,127,128]. This technique utilizes the combined effect of ultrasonic waves and agitation to induce a cavitation effect. Li et al. found that the separation efficiency increased significantly when agitation was coupled with ultrasonic treatment [128]. He et al. achieved a 99% separation using n-methyl pyrrolidone (NMP) as a solvent in conjunction with ultrasound waves [127]. Thermal treatment methods are also widely used for effective detachment of cathode from aluminium foil by high-temperature degradation of organic binder [129,130,131,132]. The temperature range for effective pyrolysis was recorded as 500–600 °C, however, vacuum was applied to avoid high-temperature aluminium brittleness [132]. Although the thermal treatment method has proven to be highly productive in terms of operational efficiency and high throughput, it has a disadvantage of producing hazardous gases due to high-temperature decomposition reactions. To avoid high energy consumption, facile mechanical separation methods including crushing, grinding, sieving and magnetic separation have been reported [133]. Shin et al. concluded that the separation efficiency of targeted metals can be enhanced by integrating mechanical methods before the metal-leaching process [134].

Both mechanical and thermal treatment methods have the advantages of being straightforward and convenient, however, suffer from producing hazardous chemicals (Table 1) [135]. Even though most pre-treatment processes have successfully been applied in different industries across the world, there are still great developments to be researched to improve the process. Such research should include methods that are not only economically feasible, but simultaneously reduce the environmental footprints.

#### 4.1.2. Metal Extraction Process

Metal extraction is the most significant part of the LIBs recycling process. In the recent decade, hydro-metallurgy, pyro-metallurgy, bio-metallurgy, and hybrid processes are widely used in industries not only for the recycling and refining of lithium but also for the extraction of other metals including nickel, copper, cobalt, and aluminium. In this section, the above-mentioned metal-extraction techniques are reviewed in terms of their strengths and weaknesses within current recycling processes.

##### Pyro-Metallurgy Processes

Pyro-metallurgical processes work on the principle of high-temperature smelting, typically in the presence of a reducing agent (e.g., coke) [135]. Normally, these processes do not require pre-treatment and the spent LIBs are directly added to the smelting furnace where they are heated beyond their melting point. Consequently, reducing the amount of carbon by converting it into alloys. The majority of the energy for the burning is provided by the combustion of the carbonaceous compounds, plastics and other volatile matter already present inside the spent LIBs. This high-temperature reductive alloy formation is followed by a secondary recovery stage through leaching. This is typically achieved with various reagents such as water or various acids (e.g., sulphuric acid (H_2_SO_4_)) [136]. Finally, solvent extraction is employed to obtain the products containing Ni, Fe, Co, and Mn. The drawback of this recovery process is the loss of lithium due to slag formation [109]. Georgi-Maschler et al. improved lithium recovery from slag by applying secondary leaching using sulphuric acid (H_2_SO_4_) [40]. In another study, Hu et al. proposed a series of steps for enhanced lithium recovery from LIBs. The method starts with the roasting of LIBs under an argon environment followed by a water leaching process to extract Li_2_CO_3_ alongside other metal components. The mixture is then subjected to CO_2_ to convert Li_2_CO_3_ to LiHCO_3_. Finally, the lithium is recovered through evaporation crystallization [137]. Träger et al. studied lithium recovery through evaporation at a temperature beyond 1400 °C, however it proved to be economically inviable due to high demand for energy consumption [138].

Although lithium recovery from LIBs using pyro-metallurgical processes is simple, they have obvious disadvantages such as high operational cost, lithium losses and risk of secondary pollution [45]. To mitigate the operational hazards, current research has been focused on either process refinement or hybrid methodologies, e.g., pyro-metallurgy coupled with hydro-metallurgy [2].

##### Hydro-Metallurgy Processes

Similar to pyro-metallurgy, hydro-metallurgical processes typically initiate with LIB pre-treatment followed by leaching, precipitation and solvent extraction. The effectiveness of a leaching process mainly depends upon the process parameters, including the type and concentration of the leaching reagent, process temperature, time duration, solid/liquid ratio and type of reducing agent [45,109]. The most commonly used leaching reagents include organic acids (ascorbic acid [139,140,141], acetic acid [142,143], oxalic acid [144,145], citric acid [141,142,146,147], tartaric acid [148] and succinic acid [149]), inorganic acids (sulphuric acid (H_2_SO_4_) [150,151,152], hydrochloric acid (HCl) [153,154,155,156], phosphoric acid (H_3_PO_4_) [157,158], and nitric acid (HNO_3_) [159]), and/or alkaline solutions to leach the desired component out for further purification [31].

Joulié et al. studied different inorganic acids including HCl, H_2_SO_4_, and HNO_3_ for lithium-nickel-cobalt-aluminium oxide (NCA) cathodes and compared their leaching performance [155]. They found that the rate of leaching was significantly higher for HCl due to the formation of chloride ions as a result of the reaction between HCl and LiCoO_2_. 4 mol L^−1^ of acidic concentration, 4 h of leaching time and 50 g L^−1^ of S/L ratio were found to be the optimum leaching conditions, obtaining almost 100% of dissolution for desired elemental recovery. In a study involving HNO_3_, Lee and Rhee et al. observed a lithium recovery rate as high as 99% when introducing H_2_O_2_ as a reducing agent [158]. Despite the high lithium leaching rate using inorganic acids, one of the major drawbacks is the production of hazardous waste (such as wastewater, Cl_2_, NO_x_, and SO_2_) that causes serious threats to environmental regulations.

In recent years, organic acids which are degradable and more environmentally friendly have been extensively studied. Such materials have shown a great potential to maintain promising Li recovery rates in hydro-metallurgical methods. Therefore, they have been widely used as alternatives to replace traditional inorganic acids. For example, Li et al. found ascorbic acid was quite effective in Li recycling from LIBs, and a lithium recovery rate of 98.5% was readily obtained [126]. Chen et al. studied the effect of citric acid in a similar process and achieved a Li recovery rate of ~99% [146]. In another study, Zhang et al. combined the biodegradable trichloroacetic acid (TCA) with a reducing agent (H_2_O_2_) and observed a Li recovery rate as high as 99.7% [160].

Irrespective of process complexity, hydro-metallurgical processes are considered to be the most favourable processes owing to their high metal recovery rate and good product quality [43].

##### Bio-Metallurgy Processes

In comparison to pyro-metallurgy and hydro-metallurgy, bio-metallurgy processes have proven to be more efficient in terms of equipment and operating costs [45]. These processes mainly rely on the in-situ production of organic and inorganic acids from microbial activities [21]. Xin et al. found that the rate of release of H_2_SO_4_ from micro-organisms significantly influenced the rate of lithium recovery [161]. Mishra et al. explored the significance of ferrous ions and elemental sulphur-oxidizing bacteria in yielding metabolites such as ferric ions and sulphuric acid inside the leaching medium, respectively. These metabolites later helped in dissolving the metal ions from the solution, including Li and Co [43]. In another study, Xin et al. found that the Li ions can be extracted through a non-contact mechanism with a maximum extraction efficiency achieved at lower system pH [162].

Compared to other Li extraction methods, bio-metallurgical processes favour mild reaction conditions are very cost-effective and simple in recovery procedures. However, the whole recovery process is time-consuming and cultivation of the desired batch of micro-organisms is difficult (Table 1) [45].

##### Other Processes for Lithium Recovery from LIBs

With the aim to develop environmentally friendly recovery processes, mechanochemical method, a hybrid process that utilizes mechanical energy to influence the physicochemical and structural properties of the metal component, has been reported [163,164,165,166]. Saeki et al. studied the effect of grinding on lithium recovery. In this method, polyvinyl chloride (PVC) was mixed with lithium-containing LIB waste (LiCoO_2_) and ground in a ball mill [163]. LiCoO_2_ decomposed in the presence of externally applied mechanical energy and converted to lithium and cobalt chlorides, while chlorine in PVC converted to its inorganic chlorides. In a later phase, these lithium and cobalt chlorides were leached out using water at an overall recovery efficiency of 100% and 90%, respectively. In a similar study reported by Wang et al., zero-valent Fe was added as a third component inside the ball mill along with PVC and LiCoO_2_. Their research achieved a Li and organic Cl recovery up to 100% and 96.4%, respectively [166]. Maschler et al. reported a hybrid process for efficient recovery of both lithium and cobalt by incorporating pyro- and hydro-metallurgy with a mechanical pre-treatment process [40]. Whereas Gupta et al. introduced a ‘chemical extraction technique’ that utilized the oxidizing properties of –Cl_2_, I_2_, and Br_2_ for fast lithium recovery from LiCoO_2_, although this method requires harsher recycling conditions [167].

Overall, conventional techniques for extracting lithium from lithium-ion batteries have many advantages. Despite this, such techniques have exhibited disadvantages such as high energy consumption, large waste production and excessive operational requirements (Table 1). Overcoming these challenges and achieving equivalent purity is crucial for future research in this field, with some research previously investigated regarding membrane technologies (Table 3).

### 4.2. Membrane Processes

Supported liquid membranes (SLM) have been studied for liquid-phase metal ion extraction/separation [168,169]. It has been considered as an alternative to conventional solvent extraction due to its advantages such as operational simplicity, low solvent demand, low energy consumption, zero effluent discharge and high selectivity [169]. Furthermore, the process is considered “green” as few chemicals are involved. In contrast with the traditional solvent extraction method, it requires much less organic solvent solely as a molecule carrier.

Swain et al. studied the separation factor of Co(II) and Li(I) from dilute aqueous sulphate media using SLM, a hydrophobic PVDF membrane with 0.45 µm pore size was used as the solid support [169]. The liquid phase was a mixture of Cyanex 272 and DP-8R, which acted as mobile carriers. Parameters such as pH, extractant concentration, feed concentration and stirring speed were studied. The group found optimal performance was achieved at pH = 5, a mixture of Cyanex 272 and DP-8R at a concentration of 750 mol m^−3^ and 350 rpm stirring speed. The resultant conditions allowed for a separation factor of Co(II)/Li(I) = 497:1. Using similar conditions, hollow fibre (HF) supported liquid membranes can be combined with non-dispersive solvent extraction (NDSX). The best condition for separation using this technique was achieved in aqueous feed at pH 6 and 750 mol m^−3^ of Cyanex 272 in the membrane. Complete separation of Co(II) and Li(I) with a 99.99% purity was achieved using the HF supported liquid membrane process with Cyanex 272 as an extractant [170]. Recently, a novel type of liquid membrane—polymer inclusion membrane (PIM) has attracted much attention due to its obvious advantages such as smaller quantities of the extractant and reduced environmental impact. PIMs have also been found to maintain high selectivity and separation efficiency compared with solvent extraction [171]. Further studies suggest that Co(II) ions were effectively removed from the source phase through the PIM containing 32 wt.% Triisooctylamine, 22 wt.% cellulose triacetate and 46 wt.% o-nitrophenyl octyl ether, with deionized water as the receiving phase [172]. Other PIM systems containing both thenoyltrifluoroacetone and trioctylphosphine oxide as the carrier and cellulose triacetate as the base polymer exhibited high selectivity of Li(I) over Na(I) and K(I) with a separation factor of 54.25 and 50.60, respectively [170].

These liquid membranes combine the benefits of both solvent extraction and membrane technologies as well as low energy consumption and low waste discharge. However, the liquid membrane typically has low stability and faces some scaling up challenges.

Membrane technologies are considered novel methods for aqueous phase lithium recovery and have been widely studied in the recent decade. However, only a few processes, such as NF and membrane distillation crystallization, have been applied at an industrial scale. Though membrane technologies have been facing some drawbacks such as membrane fouling, defects and industrial scaling-up challenges, they provide great solutions for highly efficient and environmentally friendly lithium recovery from the liquid phase.

## 5. Conclusions and Future Outlook

Lithium has quickly gained the status of being the vital building block of greener energy storage systems in the recent decade [110]. Lithium utilization in lithium batteries (LIBs), electric car batteries, energy storage grid systems, and related industrial manufacturing processes has grown exponentially over the past few years, causing major concern for the global community in terms of its prompt availability. However, the conventional technologies available for lithium extraction are either energy-intensive or time-consuming. Additionally, the extensive chemical usage makes these processes environmentally unfriendly. As a result, the development of new Li extraction methods as well as the evolution of old technologies is gaining tremendous attention worldwide. As discussed in this review, membrane technologies have successfully been attempted for lithium extraction and recycling from seawater brines and LIBs, respectively. Lithium harvesting using nanostructured membranes have the advantages of low operational cost, excellent separation efficiency, selectivity, and permeability. Furthermore, membranes result in a more environmentally friendly separation procedure. Despite these benefits, membrane technologies have succumbed to their own disadvantages. Among the disadvantages of membrane technologies are membrane fouling, membrane lifetime, and challenges for scaling up operations. Furthermore, stability has proven a great challenge, requiring future development in order to overcome poor hydro and chemical stability of membranes. Researchers are working to thoroughly address the shortcomings of these novel membrane technologies in improving its structural stability and industrial scalability. Furthermore, the optimization of existing processes and designing new membranes with improved selectivity and stability has gained much attention. The incorporation of nanofillers such as MOF materials that have tuneable framework architecture and chemical tunability can be further explored as they provide rich opportunities for creating an internal continuous ion-transport channel.

To improve lithium selectivity, a thorough understanding of the extraction mechanism through model development is required. Further, the interaction of lithium ions with different membrane support materials must be investigated. Relevant models must also be developed to visualize the internal pore structures of different membrane supports and incorporate the lithium-ion diffusion characteristics to help and improve the lithium permeability. The dynamic membrane fouling behaviour should be investigated and suitable anti-fouling agents must be designed to prevent fouling in a continuous operation. Different structural modules can be generated to improve the process scalability while maintaining process optimization.

In this work, we reviewed and compared methodologies developed recently for lithium extraction and recycling from the most abundant primary and secondary lithium resources (continental bines and LIBs), and also shared our prospects of using membrane technology as a promising alternative to replace conventional methods.

## Figures and Tables

**Figure 1 membranes-12-00373-f001:**
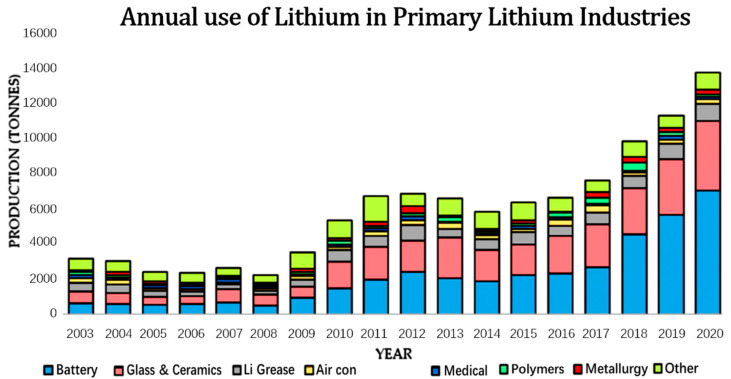
Annual use of lithium in tonnes in each of the primary lithium usage industries from 2003–2010 [9].

**Figure 2 membranes-12-00373-f002:**
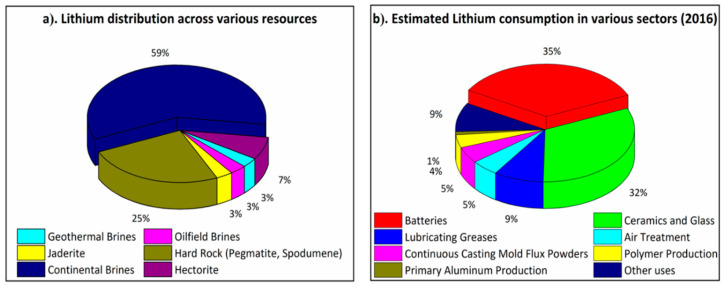
(**a**) Lithium distribution based upon primary lithium resources and (**b**) distribution of global lithium consumption for various applications.

**Figure 3 membranes-12-00373-f003:**
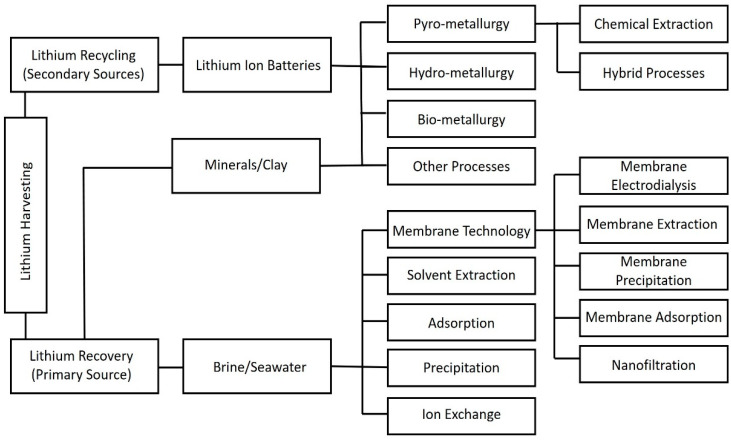
A schematic diagram summarizing the commonly used techniques for lithium harvesting.

**Figure 4 membranes-12-00373-f004:**
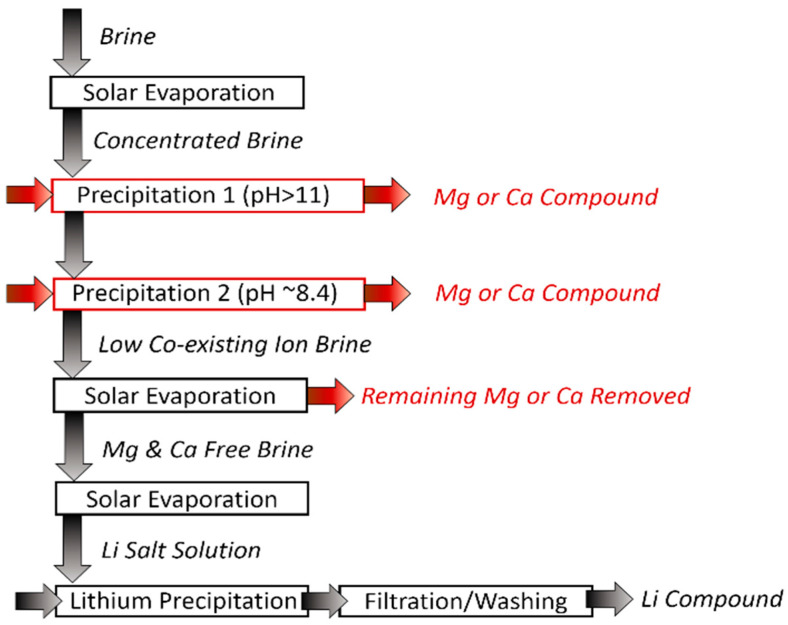
Precipitation schematic diagram.

**Figure 5 membranes-12-00373-f005:**
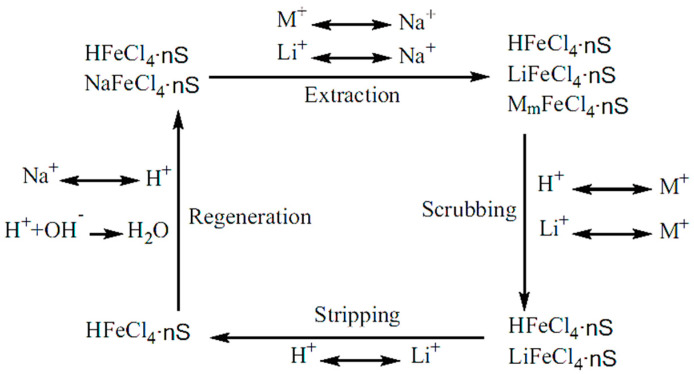
Solvent extraction process principle [55]. M refers to Na^+^, K^+^, Mg^2+^, Ca^2+^ and S refers to extractants.

**Figure 6 membranes-12-00373-f006:**
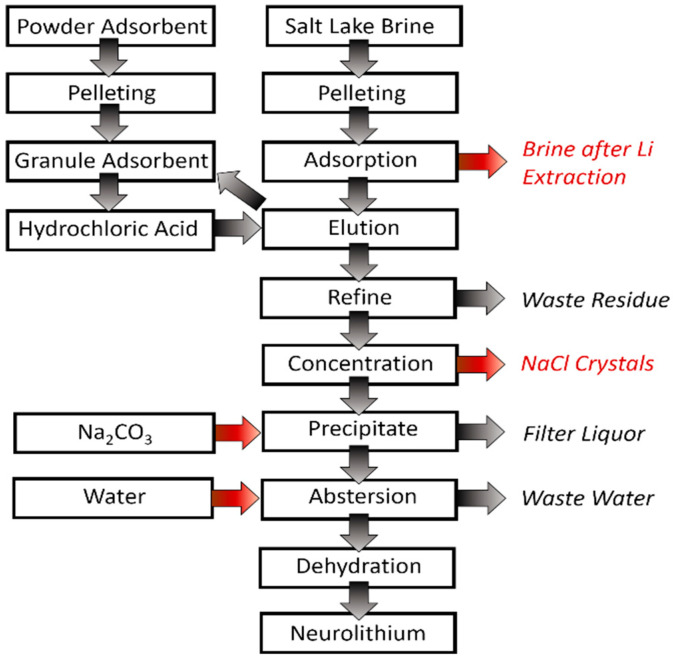
Process flow chart of lithium extraction by adsorption.

**Figure 7 membranes-12-00373-f007:**
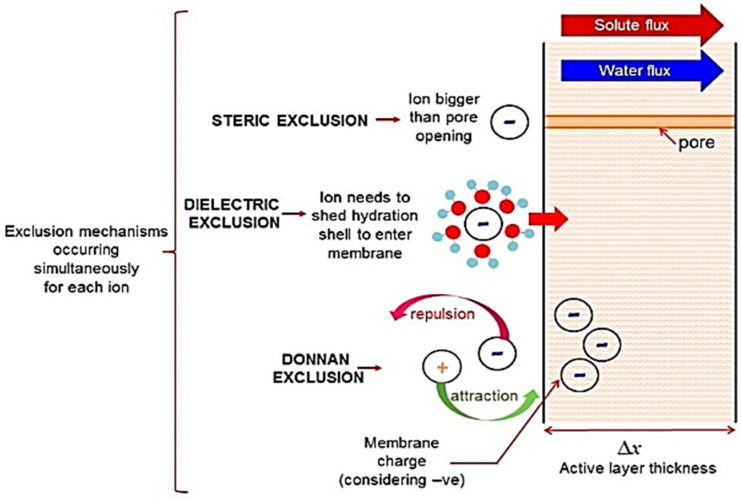
Schematic representation of solute exclusion mechanisms in nanofiltration [75].

**Figure 8 membranes-12-00373-f008:**
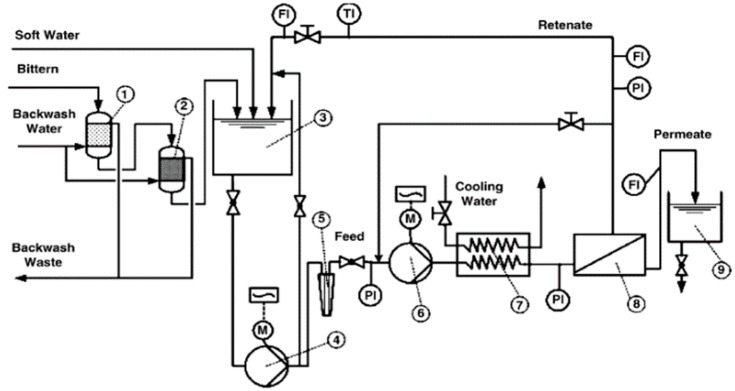
Experimental set-up of the NF treatment; (1) multimedia filter; (2) manganese dioxide sand filter; (3) feed tank; (4) feed pump; (5) polypropylene microfilter; (6) high-pressure pump; (7) heat exchanger; (8) NF membrane element; (9) permeate tank [84].

**Figure 9 membranes-12-00373-f009:**
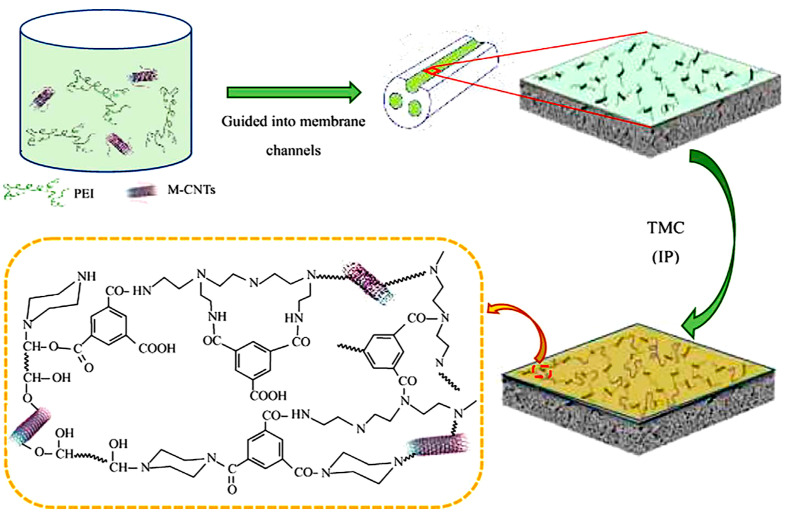
Schematic fabrication process of the positively charged NF membrane via interfacial polymerization with PEI as the aqueous precursor [86].

**Figure 10 membranes-12-00373-f010:**
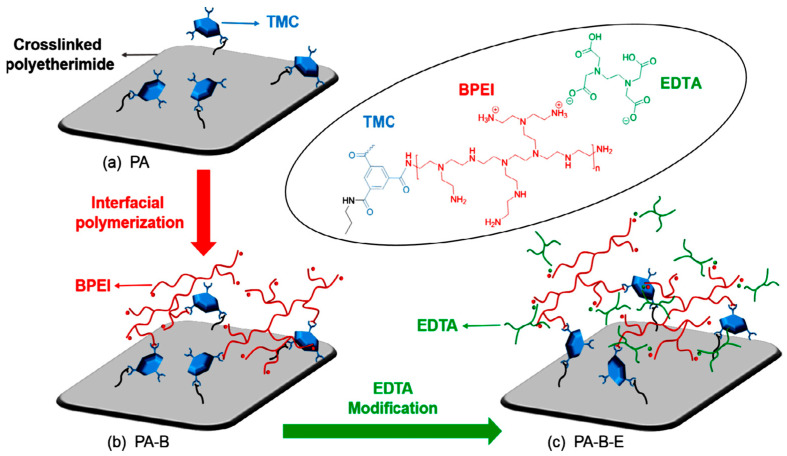
(**a**) PA membrane obtained from cross-linked polyetherimide support via interfacial polymerization between amine groups on the top layer and TMC; (**b**) PA-B membrane obtained via interfacial polymerization with BPEI; (**c**) PA-B-E membrane obtained via EDTA-modification [87].

**Figure 11 membranes-12-00373-f011:**
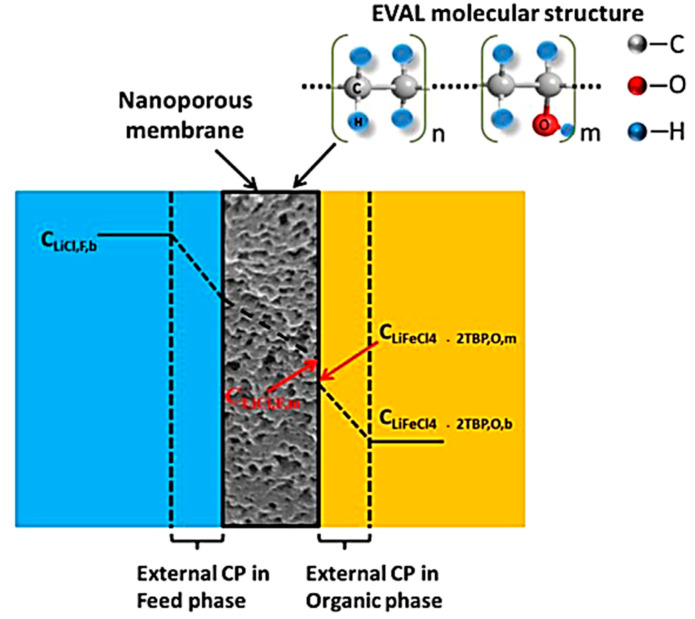
Concentration profiles of Li^+^ and/or Li^+^ complex in the membrane extraction process. “CP”: concentration polarization; “F”: Feed phase; “O”: organic phase; “m”: membrane; “b”: bulk phase [90].

**Figure 12 membranes-12-00373-f012:**
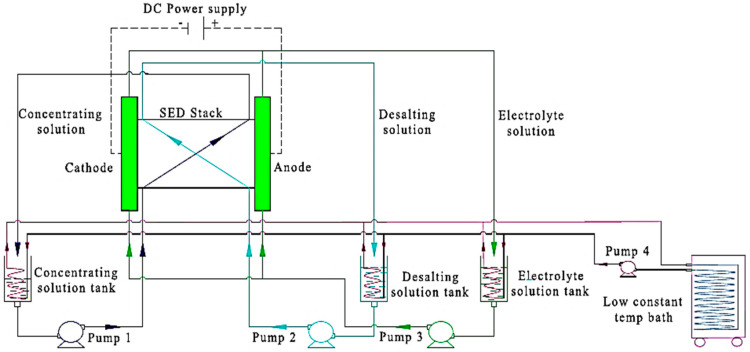
Schematic of a typical selective electrodialysis (S-ED) setup [96].

**Figure 13 membranes-12-00373-f013:**
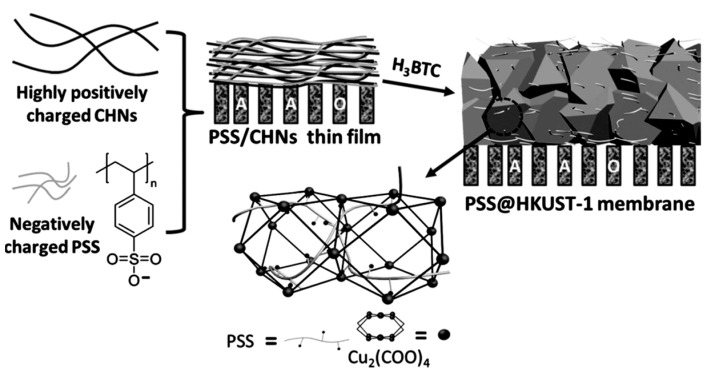
Preparation of PSS-threaded HKUST-1 membranes. CHNs = copper hydroxide nano- strands. AAO = anodic alumina, the gray bars are the anodic alumina oxide membrane [105].

**Figure 14 membranes-12-00373-f014:**
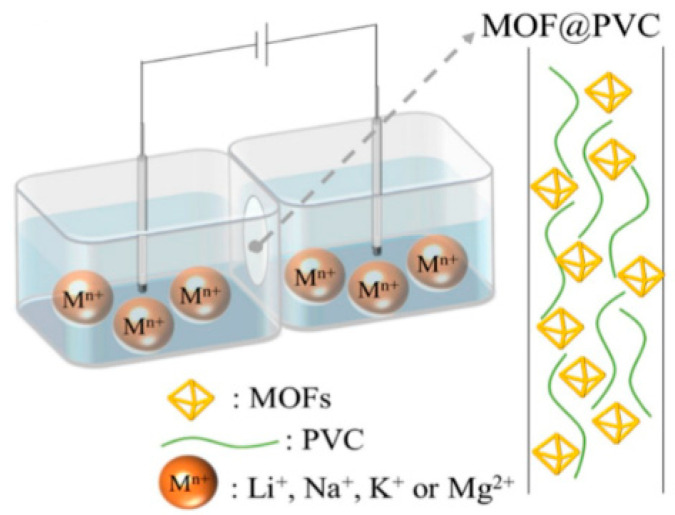
Schematic diagram of the test for ion conductivity [106].

**Figure 15 membranes-12-00373-f015:**
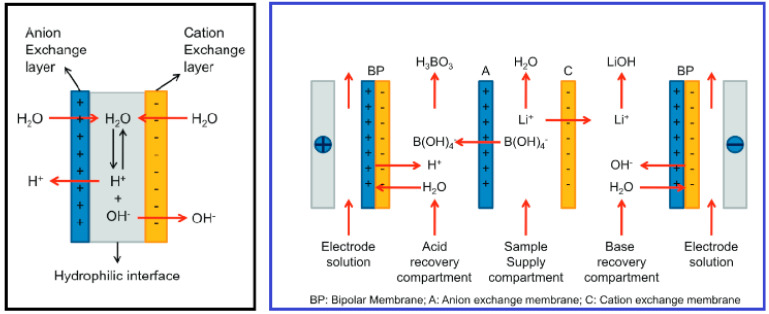
Schematic Diagram of Bipolar and Ion−exchange Membrane for Lithium and Boron Harvesting [95].

**Figure 16 membranes-12-00373-f016:**
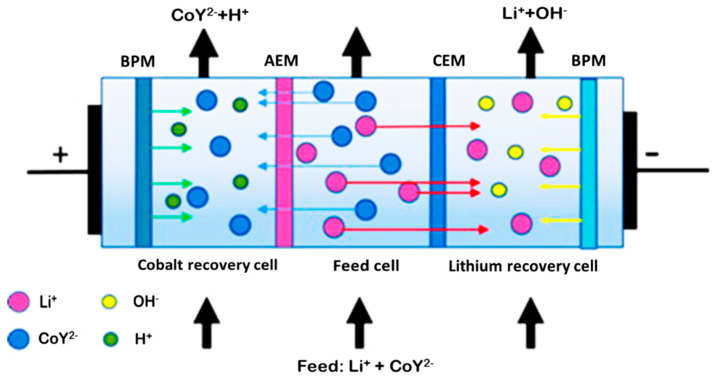
Principle of method for separation of cobalt and lithium based on electrodialysis [42].

**Table 1 membranes-12-00373-t001:** Summary of strengths and weaknesses of using old conventional methodologies for Li harvesting from Sea-water brines and LIBs.

Techniques/Processes	Strengths	Weaknesses
**Conventional technologies for Lithium extraction from Brines/Seawater**		
Precipitation	Simple Process, Green energy source (solar evaporation)	Time-consuming, A high volume of waste
Solvent Extraction	simple, adaptable and continuous operation	A high volume of waste, expensive co-agents, highly corrosive solvents, Toxic material formations
Adsorption	Simple operation, low energy consumption. Adaptable	Time-consuming, adsorbents are expensive, powdery and easily degrade in acid-driven desorption
Electrodialysis	Tailorable for Li production	Time-consuming, hazardous and corrosive materials
**Pre-treatment technologies for Lithium recycling from spent Lithium-Ion Batteries**		
Solvent dissolution	High separation efficiency	High cost of solvent, environmental hazards
Ultrasonic-assisted separation	Simple operation, almost no exhaust emission	Noise pollution, high device investment
Thermal Treatment	Simple operation, high throughput	High energy consumption, high device investment, poisonous gas emission
**Conventional technologies for Lithium recycling from Lithium-Ion Batteries**		
Pyro-metallurgy, e.g., High-temperature alloy reduction followed by Li extraction	Great capacity, simple operation	High temperature, high energy consumption, low metal recovery rate
Hydro-metallurgy, e.g., leaching and solvent extraction.	Low energy consumption, high metal recovery rate	A long recovery process, high chemical reagents consumption
Bio-metallurgy, e.g., microorganism cultivation.	Low energy consumption, mild operating conditions	Long reaction period, bacteria are difficult to cultivate

**Table 2 membranes-12-00373-t002:** A comparison of the process efficiency and percentage lithium removal from conventional methodologies.

Lithium Extraction Technologies	Process Efficiency	Percentage Lithium Removal	References
Precipitation	>90	90–99	[48,49,50,51,52]
Solvent Extraction	60–90	85–97	[53,54,55,56,57,58,59,60,61,62,63,64,65,66]
Adsorption	>75	95–99	[67,68,69,70,71,72,73,74]
Membranes	>90	80–99	[42,75,76,77,78,79,80,81,82,83,84,85,86,87,88,89,90,91,92,93,94,95,96,97,98,99,100,101,102,103,104,105,106,107,108]

**Table 3 membranes-12-00373-t003:** A comparison of the process efficiency and percentage lithium recovery in lithium ion battery based extractions.

Lithium Extraction Technologies	Process Efficiency	Percentage Lithium Removal	References
Pyro-metallurgy	>95	85–96	[109,136,137,138]
Hydro-metallurgy	>90	90–99.7	[139,140,141,142,143,144,145,146,147,148,149,150,151,152,153,154,155,156,157,158,159,160]
Bio-metallurgy	>95	~98	[161,162]
Membranes	>90	80–99.99	[168,169,170,171,172]

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
