# Peer review of "Lithium Harvesting from the Most Abundant Primary and Secondary Sources: A Comparative Study on Conventional and Membrane Technologies"

_membranes, 2022, doi:10.3390/membranes12040373_

Round 1

Reviewer 1 Report

Overall, the manuscript is in good shape. It covers an extense body of reserach focusing on Li recovery, separation, and purification methods. I recommend minor modifications prior its publication. 

There are some technicallities that need to be addressed. Please see attached pdf document with the highlighted comments for your reference.  Aswell, a few simple rearrangements on the outline. Particularly on section 3.2.

I consider section 3 and 4 could be combined so the authors dont have to include a rational on why they aren't including extraction methods from Li ores. Most of the information included could be applicable for Li beneficiation from ores anyways. As example, Section 4 has already enough redundancy with Section 3. The main difference between these sections remains on the source of Li. Combination of Section 3 and 4 will provide to the readers an easy access to Li harvesting methods. This however, remains on the discression of the authors as the information is already available. 

English language is fair, although I recommend to use technical jargon rather than common expressions. Please revise through the manuscript. Several sentences could be simplified for easier undrestanding. Not all long sentences were highlighted. 

For future submissions, please include line numbers for easier identification while commenting on the manuscript

Cheers

Reviewer 2 Report

The manuscript presents a wide review of the methods used for the recovery of lithium from primary and secondary resources.  The authors’ goal was to show some pros and contras of this method and, what can be read in the concluding remarks, to point to membrane separation as an efficient alternative to the conventional methods.  However, to make such a conclusion more convincing it is a need to show a few examples of their use at the industrial level (of course, if such systems are running).  The second comment is related to the description of SLM in the 3.2.2 section.  The authors wrote ‘ These membranes, also known as membrane contactors,….’. They continued that part by the following sentence  ‘   Song et al. investigated the use of polyethersulfone (PES) and sulfonated poly-phenyl ether ketone (SPPESK) as a binding agent to stabilize the membrane and found it stable in the presence of TBP – previously stated as a commonly used extracting agent in liquid-liquid extraction’. First of all, the membrane contactors are not a synonym of the supported liquid membrane. The first object separates two different phases while in the second one the separating phase is immobilized in the membrane (in pores for SLM or in the polymer matrix like for PIM).  The second sentence in the 3.2.2  section needs rephrasing also. The original version suggests PES and SPPESK be the binding agents. These polymers form the matrices only. The real binding agent (or carrier) is TBP.

Reviewer 3 Report

This review is interesting and timely, given recent interest in lithium recovery.  The paper provides a broad but not deep review of processes, which is fine - the field would benefit from a roadmap to guide new researchers seeking to provide context for their work.

There were some problems.  For instance, the discussion of ionic liquids on page 5 was somewhat weak and not entirely consistent with current thinking in the field.  Ionic liquids are taken to be ionic materials with melting points as high as 100 degrees Celsius, not just at room temperature.  Also, while PF6- is used in LIB electrolytes, it has been out of favor with the ionic liquids community for over a decade and so the hydrolysis issue is probably not worth raising in this context (though the authors might want to discuss it in LIB recovery).  Equation 6 could therefore be dropped or moved elsewhere.  Probably a more generic discussion of the use of functionalized ILs for metal extraction would probably be appropriate.  There are a number of reviews on the topic.

The discussion of charge-based exclusion mechanisms on page 7 could be expanded a bit, and perhaps include a schematic diagram.  I am familiar with the phenomenon, but I suspect someone who was not would have trouble following the discussion in ts current form.

The English language was generally reasonably good, but seemed to suddenly become much weaker, often changing in quality from paragraph to paragraph.  For example, on page 4, the paragraph beginning "A high Mg/Li ratio has been shown to have a sinister effect..." seems very clumsy and could stand a rewrite by a native speaker.  For instance,  "sinister" is probably not the right word, maybe "deleterious" or "problematic".

In places the language moved from being weak to being confusing and undermining the paper.  For instance, on page 5, the authors refer to "synergists," and it is not clear what the authors mean.  Synergistic solvents?  Extraction agents?  This needs to be explained.  Likewise, also on page 5,  the manuscript reads "due to the conjugation of benzene, P=O,..."  but I suspect they mean "coordination" in place of "conjugation".  There may be other places where people would find the language confusing.  It needs review not simply by a native speaker, but ideally by one with extensive knowledge in the field who can spot such problems.

A couple of specific problems:
Page 4, Equation 4, should be 2 Li+ and Li2CO3 on products side (stoichiometry is off in current version)
Page 4, Equation 5 - CaMgCO3 appears in the equation, something is not right, not sure what they're trying to do here.  Also SO4- does not exist, I think they mean sulfate ion.  Correct this equation, something is very wrong.
